# Variation in pathogen load and the pathogen load–infectiousness relationship broaden avian malaria's distribution

Christa M. Seidl [1,5] ✉, Katy L. Parise[2], Isaiah J. Ipsaro[1], Sarah Leach[1,7], Delson Hays[1,7], Ranger Morimoto[1,7], Kelsey Banister[2], Francisco C. Ferreira [3,6], Elizabeth C. Abraham[4], Jeffrey T. Foster [2], Eben H. Paxton[4] & A. Marm Kilpatrick[1]

Two aspects of host infectiousness shape pathogen transmission and distribution but are underappreciated: the relationship between pathogen load and infectiousness, and variability in pathogen load within species. We quantified the relationship between host pathogen load (parasitemia) for avian malaria (*Plasmodium relictum*) and infectiousness for biting *Culex quinquefasciatus* mosquitoes with experimental infections in canaries (*Serinus canaria*). Using this relationship, we estimated the infectiousness of 17 bird species in 11 communities in Hawaiʻi and quantified the relative contributions of infection stage (acute versus chronic) to transmission. We show that infectiousness to mosquitoes increased with parasitemia, temperature, and time since feeding. The relationship's gradual (low) parasitemia slope resulted in a wide range of parasitemias being partly infectious, and high within-host species variability in parasitemia led to extensive overlap in infectiousness among hosts. Disproportionate mosquito host utilization (inferred from relative infection prevalence) elevated the importance of a few host species, yet broad overlap in species infectiousness resulted in similar total infectiousness across most bird communities. This similarity likely contributed to avian malaria's widespread distribution throughout Hawaiʻi despite diverse host community assemblages. Our findings highlight the importance of both the shape of the pathogen load–infectiousness relationship and within-species variability in determining a pathogen's host range, transmission intensity, and spatial spread.

Some pathogens are specialists while others have broad taxonomic host ranges, and a pathogen's host range influences its geographic range[1,2]. In each location, transmission of multi-host pathogens depends on the composition of the host community, because hosts vary in abundance, infectiousness, survival, and contact rates[3,4]. Understanding how changes in host community composition affect pathogen transmission requires estimates of the infectiousness of different host species[3–5]. However, quantifying host infectiousness in

[1]Department of Ecology and Evolutionary Biology, University of California, Santa Cruz, CA, USA. [2]Pathogen and Microbiome Institute, Northern Arizona University, Flagstaff, AZ, USA. [3]Center for Vector Biology, Rutgers University, New Brunswick, NJ, USA. [4]Pacific Island Ecosystems Research Center, U.S. Geological Survey, Hawaiʻi National Park, HI, USA. [5]Present address: Maui Forest Bird Recovery Project, Pacific Cooperative Studies Unit, University of Hawaiʻi at Mānoa, Makawao, HI, USA. [6]Present address: INRAE, Ecole Nationale Vétérinaire d'Alfort, Laboratoire de Santé Animale, BIPAR, ANSES, Maisons-Alfort, France. [7]These authors contributed equally: Sarah Leach, Delson Hays, Ranger Morimoto. ✉e-mail: seidlcm@gmail.com

natural disease systems can be challenging because it entails measuring pathogen loads in wild animals and establishing a relationship between host pathogen load and infectiousness. These data can be difficult to obtain[6], and as a result, the roles of individual species in the transmission of many multi-host pathogens are unclear (but see refs. 3,4,6–8). For vector-borne multi-host pathogens, quantifying host species' importance also requires understanding host utilization of biting insects (vectors)[9]. Collectively, host pathogen loads, the pathogen-load infectiousness relationship, and vector host utilization shape how changes in host community composition will influence the spatial distribution and transmission intensity of vector-borne diseases. As anthropogenic activities such as land use change, climate change, and exotic species introductions reshape host and vector communities worldwide, improving our understanding of host infectiousness and its relationship with vectors is essential for predicting multi-host disease transmission[10–13].

The relationship between host pathogen load and vector infection (fraction of infectious vectors; Fig. 1) is a critical aspect of transmission that can vary across pathogens, strains, vector species and vector populations[14–18]. The steepness of the relationship (i.e., the slope), the location of the threshold/inflection point, and the range of pathogen load variation within host species determine both the variation and the overlap in infectiousness among species (Fig. 1). For example, when there is little within-species variation in pathogen load (Fig. 1B, narrow green and orange distributions), a relationship with a steep slope and a high threshold (Fig. 1A, black line A) will result in only hosts with higher pathogen loads infecting biting vectors; species with low pathogen loads will be dead-end, non-infectious hosts. In contrast, if the threshold is low (Fig. 1A, red line B), all high pathogen load species and most low pathogen load species will infect most vectors. Alternatively, a shallow relationship (Fig. 1A, blue line C) will result in both high and low pathogen load species being partly infectious and smaller differences in infectiousness among species. We note that a shallow

relationship can arise, in part, from heterogeneity or variation in mosquito susceptibility, which can have multiple impacts on disease dynamics[19–21]. Finally, if within-species variation in pathogen load is large (Fig. 1B, wide green and orange distributions), then individuals of both high and low pathogen load species will be partly (blue line C) or almost completely infectious (red line B or black line A). The transmission of multi-host vector-borne pathogens is, therefore, the result of interactions between multiple species' host pathogen loads with the pathogen load-infectiousness relationship. These two aspects have received relatively little study for most pathogens. Most research has focused on differences in mean pathogen loads between hosts or pathogens (West Nile and St. Louis encephalitis viruses[22–24]), which can be adequate when relationships between pathogen load and infectiousness are linear or nearly so across the range of host pathogen loads (Fig. 1, blue line C). However, doing so is problematic when relationships are nonlinear (Fig. 1, red lines A and B), which is the case for many diseases, including dengue, malaria, and many arboviral diseases[24–26]. When the relationship is nonlinear and there is variation in pathogen load within species, the mean load may be an inaccurate measure of infectiousness because, for example, a host species' mean load may be non-infectious (i.e., Fig. 1, x-axis locations where the black and red lines are at 0), while some individuals with higher than average loads may be quite infectious.

*Plasmodium relictum* is a globally-distributed and widely studied multi-host vector-borne parasite causing avian malaria, and yet it is emblematic of the uncertainty in the roles different host species play in transmission[27]. The GRW4 lineage of *P. relictum* has contributed to the decline and extinction of many bird species in Hawaiʻi[28–31], where it is the only *P. relictum* lineage and has limited genetic variation[32]. Native Hawaiian birds are hypothesized to be the primary reservoirs infecting mosquitoes with avian malaria, because it is thought that they develop higher acute and chronic parasitemias (i.e., pathogen loads) after infection than the few introduced species so far studied[29,33–37].

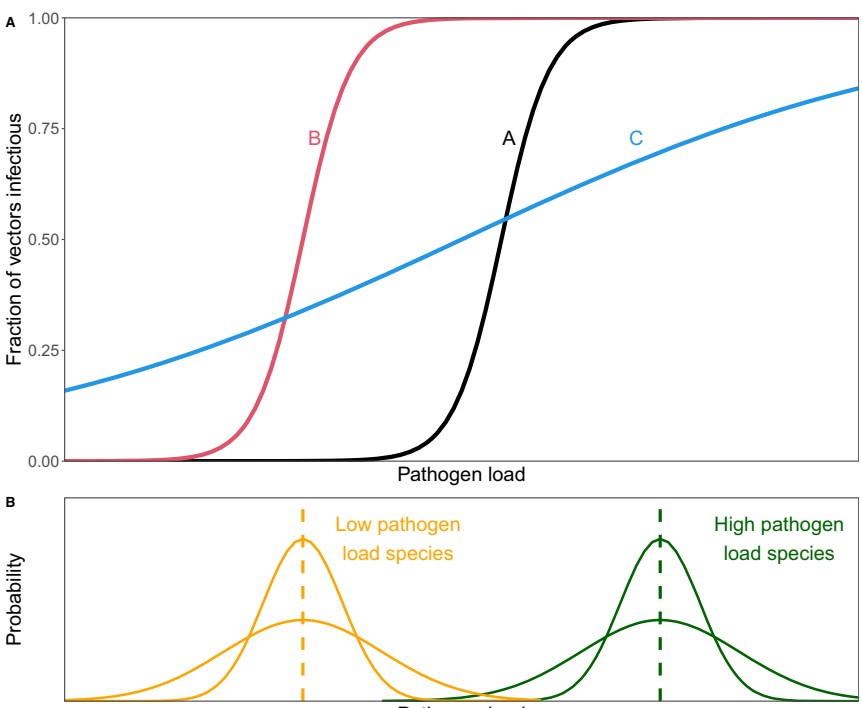

**Fig. 1 | Variation in pathogen load and the steepness of the load-infection relationship determine the infectiousness and overlap in infectiousness of host species to biting vectors. A** Three possible relationships (lines A, B, and C) between pathogen load and the fraction of infectious vectors that transmit after feeding on a host with that pathogen load. **B** Wide and narrow distributions show ranges of variation in pathogen load for two hypothetical species with higher (dark green distributions and vertical dashed line) and lower mean (dashed lines) pathogen load (orange distributions).

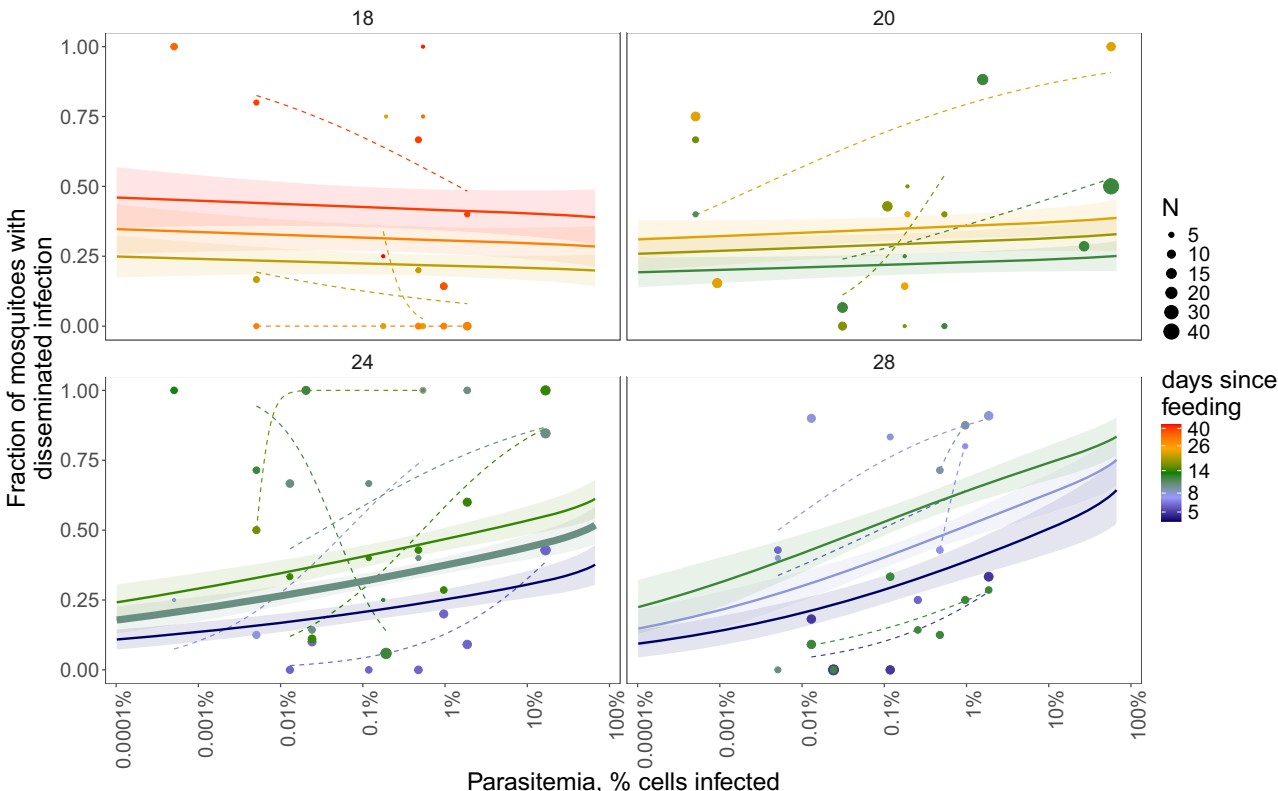

**Fig. 2 | Fraction of mosquitoes with disseminated infections as a function of** *Plasmodium relictum* **parasitemia on a log₁₀ axis for four temperatures (18, 20, 24, 28 °C) and a range of days since feeding that differed by temperature.** Points show groups of blood-fed mosquitoes tested on a single day, with the size of points indicating sample size (mean 8.2, range 4–40), and color showing the number of days since feeding. Colored lines and ribbons show the fitted model for the probability of disseminated infection and SE, which included bird parasitemia, days since feeding, temperature, and both two-way interactions (Table S1). The thicker middle line in the 24 °C panel shows the model fit for 10 d after feeding and is used for bird infectiousness calculations (Fig. 3 and Table S4). Dashed lines show individual fitted binomial regression lines to each set of points from the same day, since feeding and temperature, to illustrate the trends between parasitemia and the fraction of mosquitoes with disseminated infections.

However, more than 50 introduced bird species are established in Hawaiʻi, and avian malaria transmission occurs in communities composed entirely of introduced birds[37,38]. While parasitemias of most introduced birds are unknown, parasitemias in some species are similar to those of native birds[39]. To better understand avian malaria transmission and the contributions of different bird species in Hawaiʻi, three key components are needed. First, parasitemias (including within-species variation; Fig. 1 distributions) in most introduced bird species need to be measured. Second, the relationship between host parasitemia and infectiousness to biting mosquitoes needs to be quantified (Fig. 1 lines). Finally, mosquito feeding patterns relative to host abundance need to be determined.

In this work, our goal was to determine the role of different bird species and thereby community composition in shaping the transmission ecology of avian malaria in Hawaiʻi. In our multi-step approach, we first used laboratory experiments to quantify the relationship between three key factors: host parasitemia, temperature, and time since feeding, and infectiousness, or the fraction of infected biting Hawaiian *Culex quinquefasciatus* mosquitoes that had disseminated infections. We note that this definition of infectiousness incorporates midgut escape (a mosquito trait), and we did so because the extent of midgut escape depends on host parasitemia. Second, we compared the infectiousness and role in transmission of the acute and chronic stages of malaria infection to assess whether parasitemia data from wild-caught birds, which are likely to be chronically-infected[39], can be used to estimate species' infectiousness and role in transmission. Third, we used the parasitemia-infectiousness relationship to estimate the infectiousness of 17 bird species using parasitemia data from wild-caught birds. Fourth, we estimated relative mosquito host utilization using patterns of avian malaria infection at 78 sites. Fifth, we integrated our estimates of host infectiousness, mosquito host utilization, and bird density from 11 sites in Hawaiʻi to evaluate species-specific contributions to mosquito infection and pathogen transmission potential, $R_O$. Finally, we examined spatial patterns of avian malaria occurrence across sites with differing species composition. Here, we show that avian malaria has a very broad distribution in Hawaiʻi due in part to a gradual pathogen load-infectiousness relationship and substantial variation within species in pathogen load.

## Results

### Avian malaria pathogen load-infectiousness relationship

We fed 820 mosquitoes on canaries with 21 parasitemias over seven orders of magnitude, ranging from 0.0000051–0.573 (fraction of infected red blood cells (RBCs)) and tested their abdomens and combined thorax/head/legs (disseminated infection) for avian malaria DNA by qPCR. While there was observation error due to small-moderate sample sizes for each data point (a parasitemia-temperature-day), the probability of mosquitoes having disseminated infections, which was our measure of bird infectiousness, increased with parasitemia, temperature, days since feeding, age, and the effects of both parasitemia and days since feeding increased as temperature increased (Fig. 2 and Table S1; Cox-Snell pseudo-$R^2$ = 46%). At the warmer temperatures, the effect of parasitemia on the probability of mosquitoes having disseminated infections was strong, whereas at the coolest temperature (18 °C), the effect was undetectable (Fig. 2; compare the slope of lines in the lower right panel at 28 °C to the upper left panel at

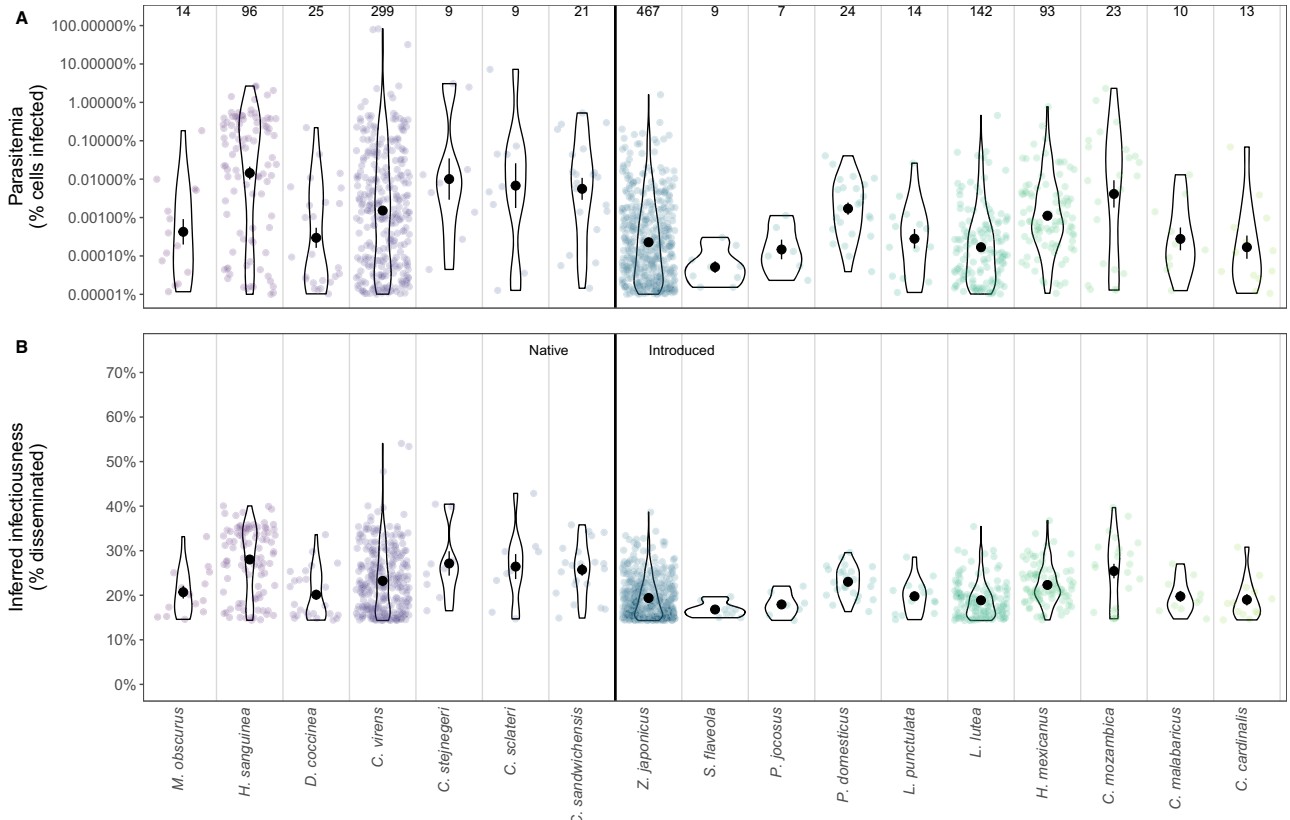

**Fig. 3 | Parasitemia and infectiousness for 17 bird species in Hawai'i (7 native species (left of the vertical black line), 10 introduced species (right of the vertical black line)). A** *Plasmodium relictum* GRW4 parasitemia on a $\log_{10}$ scale. **B** Infectiousness was estimated as the fraction of mosquitoes with disseminated infections, 10 days at 24 °C after feeding on each parasitemia estimate from **A**. In **A** and **B**, points show values for individual wild-caught birds, with the number of parasitemia values displayed above each species (total $N = 1207$). Black circles and error bars are the mean values and SE for each species. Native and introduced species differed significantly in both parasitemia and infectiousness (generalized linear mixed effects model with a beta distribution and logit link with species as a random effect: Parasitemia: Native coef. $0.27 \pm$ SE 0.091, $z = 2.97$, $P = 0.0030$, Intercept $-5.77$, random effect variance 0.0086; Infectiousness: Native coef. $0.21 \pm$ SE 0.071, $z = 2.91$, $P = 0.0036$; Intercept $-1.35$, random effect variance 0.016; see also Fig. S4).

18 °C). Although this four-dimensional analysis (with 2 two-way interactions) is somewhat complex, the increase in the probability of mosquitoes having disseminated infections with increasing parasitemia was present in almost all (13/14) groups of mosquitoes tested on the same day post-feeding and in each of the three warmer temperatures (the dashed lines in Fig. 2 for the 20, 24, and 28 °C panels). Despite the consistent positive effect of parasitemia on disseminated infections at temperatures ≥20 °C, the relationship was relatively gradual, even at the warmer 24 °C temperature: increasing parasitemia across a $10^5$ range (from $10^{-6}$ to $10^{-1}$) increased the probability of disseminated infections in mosquitoes 10 days after feeding by only ~2.4-fold, from 18.0% to 43.7% (Fig. 2, 24 °C degree panel, middle line).

Increasing temperatures also increased the effect of time on infectiousness: at 20 °C, a four day increase in days since feeding (18 to 22) only increased the probability of mosquitoes having disseminated infections by 19% (from 29% to 34% averaged across the range of parasitemias); whereas at 28 °C, a four-day interval (8 to 12) led to a 27% increase, from 39% to 49%. The patterns for abdominal infections were similar (Fig. S3 and Table S2).

**Role of host infection stage in transmission**
For two native Hawaiian birds, 'Apapane (*Himatione sanguinea*) and Hawai'i 'Amakihi (*Chlorodrepanis virens*), we found that the low parasitemia phase of infection likely results in many more disseminated mosquito infections than the high parasitemia phase (Figs. S4 and S5; Table S4). We estimated that approximately 8.54–13.8 times more infectious mosquitoes would be generated when feeding on 'Apapane

and Hawai'i 'Amakihi during their much longer (13.2–25.3 times) low parasitemia chronic stage than during their short high parasitemia acute phase. This is because the much higher parasitemias were only 1.54–1.83 times more infectious to mosquitoes (Fig. S4 and S5; Table S4).

**Parasitemias and infectiousness across bird species**
We tested 4218 blood samples from 34 bird species captured at 78 sites in the wild, and 1275 samples from 28 species tested positive for avian malaria by qPCR (Fig. 1). Although there was variation among species in parasitemia, the variation within species was enormous, and parasitemias broadly overlapped for many species (Fig. 3A; filled circles and violins). For example, parasitemias and infectiousness estimates from native 'Apapane (see Table S5 for bird English common and scientific names) and Hawai'i 'Amakihi (2.5th and 97.5th quantiles: Parasitemia $10^{-5}$–2%, Infectiousness 14–39%) broadly overlapped with those of introduced Warbling White-eyes (*Zosterops japonicus*) and House Finches (*Haemorhous mexicanus*; 2.5th and 97.5th quantiles: Parasitemia $10^{-5}$–$10^{-1}$%, Infectiousness 14–32%) (Fig. 3A, B). Nonetheless, on average, native bird species had 8.0-fold higher geometric mean parasitemias than introduced bird species (native $= 2.68 \times 10^{-3}$ %, introduced $= 3.36 \times 10^{-4}$ %; Fig. 3A and Fig. S4), and this difference resulted in a 4.2% higher probability of mosquitoes developing disseminated infections (introduced species $= 20.2\%$; native species $= 24.4\%$; Fig. 3B and Fig. S4).

Infection prevalence varied substantially among species for both native and introduced birds (Fig. S6). When accounting for site, all native birds had higher or similar infection prevalence than the

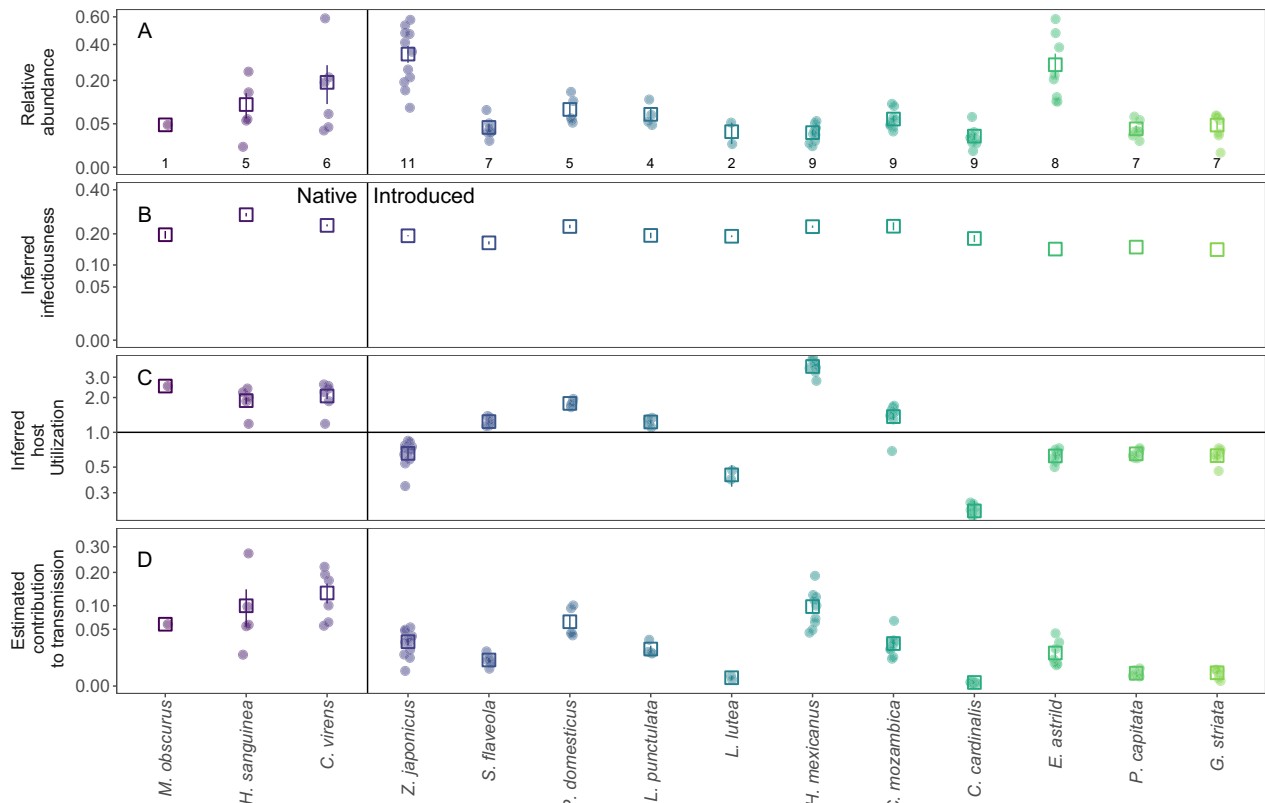

**Fig. 4 | Relative host abundance, host infectiousness, inferred vector host utilization and host contributions to transmission for 14 bird species at 11 community sites in Hawai'i (see Tables S3 for site details and S5 for full species names). A** Relative abundance of species based on point count sampling (see Fig. S7 for absolute densities). **B** Mean infectiousness (from Fig. 3B). **C** Inferred host utilizations, on a log scale, estimated using the exponentiated coefficient from the model fit to infection prevalence data (Fig. S6) and relative abundance (see Methods); the horizontal line shows a host utilization of 1 (mosquitoes feed on hosts in proportion to their abundance). **D** Estimated relative contributions of species to transmission quantified as: Inferred host utilization[2] * Relative abundance * Inferred infectiousness (see Methods). Points show values from individual sites, open squares and error bars show mean and SE across sites, except for species' Infectiousness values, which were not site specific. The y-axes in panels **A**, **B**, and **D** are on a square root transformed scale to improve interpretability. The error bars in panels **A**, **C**, and **D** show ±1 SE with sample size (# of sites) shown in panel A. The error bars in (**B**) are taken from Fig. 3B with sample sizes from Fig. 3A.

abundant, widespread, introduced Warbling White-eye. However, other introduced species had both higher (House Finch) or much lower infection prevalence (Northern Cardinal; *Cardinalis cardinalis*). Given evidence that species are equally susceptible (see Methods), differences in infection prevalence patterns suggest that mosquito host utilization varies substantially. House Finches appeared to be the most overutilized hosts, followed by four native species, while Warbling White-eye, Northern Cardinal and Red-billed Leothrix (*Leiothrix lutea*), were much less frequently fed on by mosquitoes (Fig. S6). These differences in host utilization (estimated using the differences in infection prevalence), played a key role in the importance of both native and introduced species in avian malaria transmission, as described below.

**Community composition, total infectiousness, and malaria distribution**

We recorded 28 species during point counts at 11 community sites on Hawai'i Island and estimated densities (birds/ha) for the 20 species detected multiple times (Fig. 4A; Figs. S1 and S7, Table S5). Warbling White-eye was the most abundant species, comprising 9–58% of birds at the 11 sites, and Hawai'i 'Amakihi was the most abundant of four native bird species, comprising 4–59% of all birds at the 6 sites where it was present (Fig. 4A and Fig. S7).

We detected *P. relictum* at 63 of 64 sites where we sampled >8 birds for infection prevalence and parasitemia (Fig. S8). This included many high elevation sites (>1700 m), where temperature limits

mosquito populations and *P. relictum* replication[40]. Avian malaria was detected in all 11 sites that we censused on Hawai'i Island (Fig. S9), and although these communities varied substantially in species composition (Fig. 4A and Fig. S7), their estimated community infectiousness values, or the sum of species' contribution to transmission, were similar (Figs. 4D and 5: coefficient of variation across sites: 11.8%, range: 0.265–0.371). This was due to the relatively small differences among species in infectiousness (Figs. 3B and 4B). In contrast, the highly varying estimates of mosquito host utilization (Fig. 4C), derived from relative infection prevalence (Fig. S6), substantially influenced the role of individual species in transmission and suggested that some species were much more important than others (Fig. 4D). Introduced House Finches, which were uncommon, moderately infectious, and frequently infected, were the most important species at five of the nine sites where they were present (making up an average of 30.4%, range 13–55%, of the estimated contribution to transmission or community infectiousness), despite being only 3.2% (range 1.2–5.6%) of the birds at these sites (Fig. 4A). Native 'Apapane and Hawai'i 'Amakihi were the most important species at five sites where House Finches were absent or rare (Figs. 4D and 5) and were the next most important species at most sites where House Finches were present (Figs. 4D and 5) because they were overutilized, moderately abundant, and highly infectious. In contrast, Warbling White-eyes, which were present at all sites and often highly abundant (Fig. 4A; mean 34%, range 9%–58% of all birds), likely played a minor role in transmission (Figs. 4D and 5; mean 9.4%, range 1.3–15.7%), despite being moderately infectious (Fig. 4B). This was

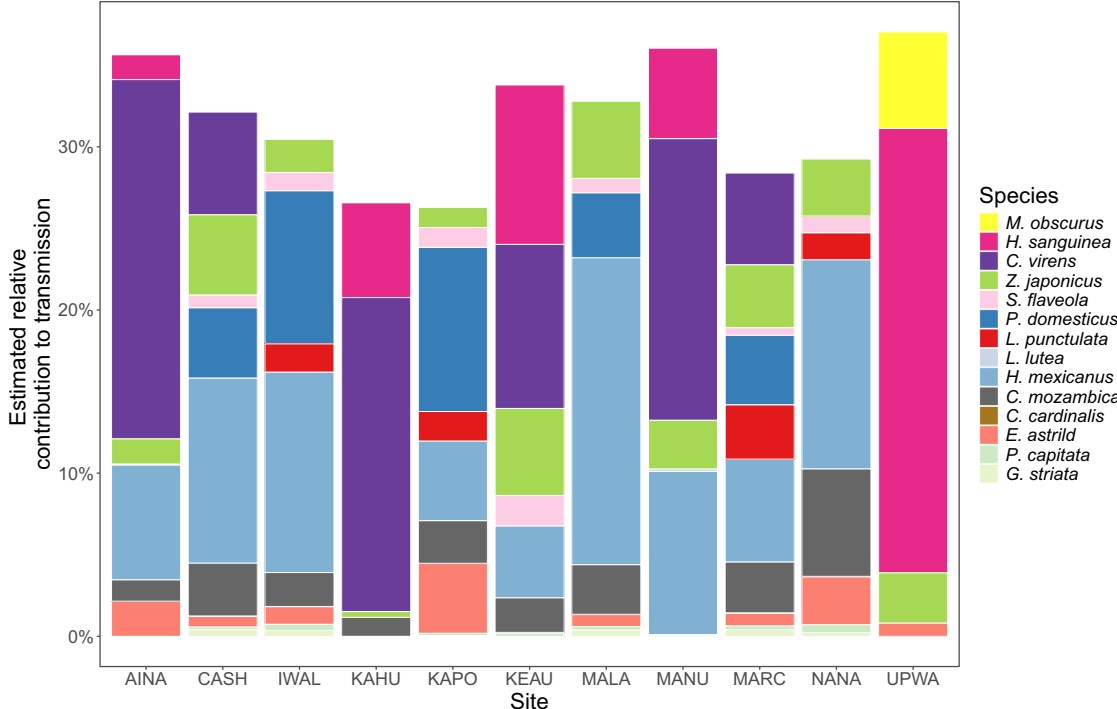

**Fig. 5 | Stacked histogram of estimated species' relative contributions to transmission at 11 community sites where we censused birds.** Estimated community infectiousness, or the sum of species contributions to transmission at each site, is the total height of each bar. Comparisons of individual species contributions are also visualized in Fig. 4D.

because they had much lower relative infection prevalence (Fig. S6), suggesting they were infrequently fed upon (Fig. 4C). It's worth noting that at most sites, only two or three species likely played an important role in transmission (Figs. 4D and 5), primarily due to the highly skewed estimates of mosquito host utilization (Fig. 4C). Host contributions to transmission were uncorrelated with species mass and did not differ between native/introduced species (generalized linear model of mean contributions to transmission with a gamma distribution and log link: Log(contribution to $R_0$) = −2.97 to 0.023 (SE 0.025) *Mass + 1.22 (SE 0.74)*Native; P(Mass) = 0.39; P(Native) = 0.14).

## Discussion

Avian malaria transmission occurs widely across the Hawaiian Islands, including at nearly all of our sites, despite highly variable host communities[37,38]. In contrast, many other pathogens are only present in a subset of host communities, often because only a few hosts are infectious enough to sustain transmission[41–43]. We found that the widespread distribution of avian malaria in Hawai'i is likely driven, at least in part, by a relatively gradual relationship between host parasitemia and infectiousness to biting mosquitoes, and by substantial within-species variation in parasitemia (Fig. 1A, line C, Fig. 1B, broad distributions). Together, these two factors produced broad overlap in the infectiousness of many native and introduced bird species to mosquitoes, allowing many species in both groups to contribute to transmission. In comparison, a relationship with a steeper slope/threshold (but an inflection point) and lower variability in parasitemia within host species would have resulted in a far more limited geographic distribution and host range for avian malaria in Hawai'i because only a few high parasitemia species would be infectious, rather than a broader set of moderately infectious host species. The role of these two factors in influencing the distribution of other vector-borne pathogens has yet to be explored, but substantial variation exists among pathogens. For example, West Nile virus, another near globally-distributed pathogen, also exhibits a relatively gradual viremia-infectiousness relationship[22], and substantial within-species

variation in viremia[23,24]. In contrast, the viremia-infectiousness relationship is quite steep for the relatively host-specific *Plasmodium falciparum*[26]. For dengue virus (DENV), the viremia-infectiousness relationship varies among serotypes: DENV2 shows a steeper relationship in *Aedes aegypti* than other serotypes[25], while in *Aedes albopictus*, DENV1 has the steepest relationship of the serotypes[44]. This variation in the slope of these relationships likely arises from a combination of variation in susceptibility among mosquitoes, the steepness of the relationship for individual mosquitoes, and variation among pathogen genotypes. Future studies could examine the effect of variation in the slope of this relationship using distribution data for DENV or other pathogens[45].

We found that a limited set of species likely play an important role in infecting mosquitoes with avian malaria, with mosquito host utilization playing a key role in determining the importance of species in transmission, as is the case for other mosquito-borne pathogens[4,9,46,47]. One introduced species with low relative abundance, the House Finch, was the most important species for infecting mosquitoes at most sites where it was present. This was because it was moderately infectious and had a higher infection prevalence and thus was inferred to be highly fed upon by mosquitoes. Two native bird species were also important when they were present at moderate abundances, due to a combination of being highly infectious and having moderate infection prevalence (and inferred attractiveness to mosquitoes). In contrast, the most abundant and widespread species, the introduced Warbling White-eye, appeared to be relatively unimportant in transmission because it had low infection prevalence, suggesting it was infrequently fed upon by mosquitoes. The role of mosquito feeding patterns in amplifying the importance of uncommon species and diminishing the role of highly abundant species in Hawai'i mirrors the disease ecology of American Robins (*Turdus migratorius*; uncommon, disproportionately fed upon) and House Sparrows (abundant, infrequently fed upon) in West Nile virus transmission[4,7,48]. More generally, evidence suggests that vectors rarely feed on hosts in proportion to their abundance and disproportionate feeding on some species can

play a key role in influencing the role of different species in transmission[9,46,47]. The importance of host utilization by vectors can be especially influential if variation among species in infectiousness is limited, as we found here.

The shallow slope of the parasitemia-infectiousness relationship and the large variation in parasitemia within species can partly explain the substantial transmission of avian malaria in bird communities in Hawai'i containing only introduced species[37,38], which were hypothesized to be non-infectious hosts[29,49]. Our findings suggest these communities are moderately infectious despite introduced species having lower mean parasitemias (Figs. 3A and 4). If these sites have higher mosquito abundance, survival, or other factors that increase transmission[37], these factors could counterbalance the slightly lower infectiousness of the host community. Our results suggest that many, and possibly most, bird communities in Hawai'i can support transmission of avian malaria. The widespread distribution of malaria in Hawai'i may prevent native birds, including many endangered or threatened species that suffer substantial mortality from infection[29,34], from persisting in or recolonizing many sites where they could otherwise exist.

We found that many more infected mosquitoes arise from chronic, low parasitemia stages of infection than from acute high parasitemia stage infections in Hawai'i 'Amakihi and 'Apapane, as is the case for other pathogens such as *Plasmodium vivax*, and filarial parasites[50,51]. Low-level parasitemias also play an important role in the transmission of human malaria *P. falciparum*, in part due to a highly variable relationship between overall parasite density and mosquito-infecting gametocyte density within hosts[26,52]. A highly variable fraction of malaria parasites being gametocytes may explain the gradual relationship we observed between total parasite density (parasitemia) and disseminated infections in mosquitoes[53]. Regardless of the mechanism, the much larger importance of the chronic phase of infection in infecting mosquitoes may reduce the need for experimental infection studies to quantify the species' role in transmission. This is fortunate because these studies are challenging to conduct, especially with threatened species. Instead, parasitemia data generated from sampling of wild animals, which are mostly chronically-infected individuals[39], can provide an initial estimate of a species' infectiousness, as we have done here.

Four limitations affected this study. First, we were unable to analyze the probability of transmission given disseminated infection, or the fraction of mosquitoes with positive salivary secretions[54]. We attempted to use a salivary secretion assay on anaesthetized mosquitoes, but the method sometimes produced false negative results (*i.e.*, no mosquito DNA in the salivary secretions), likely due to inadequate mosquito saliva being expectorated into the capillary tube[55]. Previous work on *P. relictum* in *Cx. quinquefasciatus* from Hawai'i found the fraction of mosquitoes with malaria oocysts in their midgut that also had vertebrate-infecting sporozoites in their salivary glands increased with temperature from 21% at 17 °C to 98% at 29 °C[40] (Fig. S10). This pattern supports the temperature-dependence we found in the parasitemia-infectiousness relationship, but this study[40] did not examine what fraction of mosquitoes had infections in their thoraxes or how the fraction with sporozoites changed with days since feeding or parasitemia. Second, we assumed our pathogen load-infectiousness relationship generated from infected domestic canaries could be used to estimate infectiousness for wild bird species using their parasitemias. One previous study found variation between two bird species in the fraction of mosquito vectors infected with West Nile virus despite similar viremias[56]; however, sample sizes in this study were small, and it's unclear how common and large this variation is in nature. Third, we estimated mosquito host utilization using relative patterns of infection prevalence, rather than identifying the source of blood-fed mosquitoes. This approach assumes that species are equally susceptible, have similar infectious periods, and that chronic

infections are equally likely to be detected in all species. If some species are less susceptible or have chronic infections that are more difficult to detect (e.g., they are below the threshold of detection by qPCR; a parasitemia of approximately $1 \times 10^{-7}$ infected RBCs[39]), our approach will underestimate mosquito host utilization for that species. However, even if there are unmeasured differences in susceptibility or undetectable infections among species (which is inconsistent with available data[29,34,35,57–60]), simulations indicate that using relative infection prevalence or directly measured feeding patterns are approximately equally accurate (Fig. S11). This is because species with undetectable infections will be poorly infectious (the qPCR threshold of 40 cycles or $1 \times 10^{-7}$ infected RBCs is <10% infectious; Fig. 3B) and species or individuals that don't become infected (i.e., aren't susceptible) won't transmit malaria to biting mosquitoes. Finally, we had no parasitemia data for several bird species that were present at our 11 community sites. We estimated the infectiousness of some of these species using evolutionarily close relatives, but for a few others, we were forced to use a broad average. Some of these species may be dead-end hosts for *P. relictum* (e.g., Phasianidae). These limitations likely add uncertainty to the precise quantitative results we found, but are unlikely to alter our broader findings.

Some pathogens are generalists with wide spatial distribution, whereas others are limited to a small set of hosts and a narrow geographic range. Avian malaria (*P. relictum*) is a globally distributed vector-borne pathogen that infects hundreds of bird species[27]. In Hawai'i, it has contributed to multiple species extinctions and continues to limit the distribution of several endangered species[28–30,61]. Our results suggest that large within-species variability in pathogen load, coupled with a gradual relationship between pathogen load and infectiousness, increases the number of host species contributing to transmission: both individual host species and whole communities had broadly overlapping infectiousness. More generally, the steepness of the relationship between pathogen load and infectiousness and variation in pathogen load within species likely influence not only which species can act as competent hosts, but also the spatial extent of pathogen transmission. These factors may therefore play a central role in shaping the host range and geographic distribution of many vector-borne diseases.

## Methods

### Avian malaria and mosquito collection and maintenance

We collected three isolates of *P. relictum* (lineage GRW4) from wild birds at three sites on Hawai'i Island as previously described[62]. Past studies found limited genetic variation and no difference in pathogenicity between Hawaiian isolates of GRW4[57,63]. Isolates were passaged one to five times in canaries via intramuscular inoculations of 50–100 μl of infected fresh blood before use in mosquitoes feeding trials (see below), or before cryopreservation for future use[64]. All work with wild and laboratory birds was performed under animal care and use protocols approved by the Institutional Animal Care and Use Committee at the University of California in Santa Cruz, USA (Kilpm2003).

*Culex quinquefasciatus* mosquito egg rafts were collected from three locations on Hawai'i Island between 2020 and 2023 (Fig. S1) and shipped within 24 h to Santa Cruz, California (U.S. Veterinary Permits 139503 and 611-21-327-00722). Upon arrival, rafts were floated in plastic pans (44 cm × 25 cm × 10 cm) filled with 1 L of deionized water at 26 °C with 70–80% humidity under a 12 L:12D h photoperiod. Hatched larvae (200–350 per pan) were fed daily 0.2–0.4 g of ground fish food (Koi's Choice® Premium Fish Food), and pupae were transferred to cages (30 cm³, BugDorm) in the same incubator. Emerged adults were fed ad libitum on 10% sucrose solution-soaked cottons. Five- to 37-day-old ($\bar{x} = 20$) wild-collected, F1, and F2 generation adults were used in experimental feedings. All mosquitoes were sucrose-starved for 48 h prior to blood feeding. To create F1 and F2 generations, blood-

fed mosquitoes were provided a small cup of water for egg laying starting 2 d after blood feeding. Cups were checked daily for egg rafts.

## Experimental infection and mosquito feeding assays

To establish our infection assays for birds and mosquitoes, we inoculated canaries intramuscularly with 50–200 μL of 0.1–4.25% *P. relictum* infected whole blood. Infected blood was either fresh, containing an isolate passaged 1–4 times in other canaries, or a thawed deglycerolized sample. Starting on day 5 post-infection and every 3 d after, we took 5–10 μL of blood by brachial venipuncture and screened blood using both thin blood smears and quantitative PCR (qPCR) to detect infection and estimate parasitemia[39,65]. Once parasitemia was detectable, unrestrained canaries were placed inside a vertical PVC cylinder within a mosquito cage containing 100 female mosquitoes (Fig. S2). Birds and mosquitoes were held together overnight from 2000 to 0600 h (10 h) at 24 °C. We collected engorged mosquitoes, divided them into four groups, and transferred the groups into incubators set to 18, 20, 24, or 28 °C, temperatures which are common across a range of bird habitats in Hawai'i[66]. We held mosquitoes within each incubator until dissection. Individual infected birds could be exposed multiple nights to biting mosquitoes, but individual mosquitoes were only exposed once to an infected bird.

We dissected a subset of mosquitoes from each incubator at three time points between days 5 and 44, the points varying depending on the temperature of the incubator. We chose time points that would capture the rise in the fraction of mosquitoes with disseminated infections over time[67]. Mosquitoes were dissected by separating the thorax from the abdomen at the scutellum with a sterile dissection needle. The abdomen was placed in one 1 mL vial containing 0.5 mL of 70% ethanol, and the combined head, thorax, and legs were placed in a second vial. All samples were stored for 1–90 d and frozen at −20 °C before qPCR was used to detect *P. relictum* DNA.

## Bird sample collection and censuses

We collected blood samples to quantify *P. relictum* infection from 34 species of wild birds captured using 38-mm mesh mist-nets at 78 forested sites between 29–2000 m on the islands of Kaua'i, O'ahu, Maui, and Hawai'i Island (Hawai'i, USA) between 2015–2022[38,68]. We drew 25–100 μL of blood by brachial venipuncture and placed it in 1 mL of Queen's Lysis Buffer[69]. We stored samples at room temperature for 1–90 days or froze them at −20 °C before using qPCR to quantify parasitemia.

To characterize the composition and relative abundance of species in bird communities, we conducted unlimited-distance point counts at 11 of our sampling sites (Fig. S1). At each community site, we conducted four 6 min point counts between 0600 and 1000 h from February to June 2020, with points separated by at least 200 m. We estimated the density of each species (birds/ha) using distance sampling using the Distance package in R[70]. We used a half-normal key function, a truncation distance of 95 m, and a species-by-site detection function (which fit better than just species-specific detection functions; ΔAIC = 180.6); the model fit was very good ("badness" of fit: $P = 0.82$). We conducted fieldwork under Hawai'i Division of Forestry and Wildlife Protected Wildlife Permits (WL19-23; WL 17-08), USGS Bird Banding Laboratory permit numbers (#23600, #21144), and a Hawai'i State Access and Forest Reserve Special Use Permit.

## qPCR analysis

We extracted DNA from avian blood and mosquito samples using a Qiagen DNeasy Blood & Tissue kit (Qiagen, Hilden, Germany) following the manufacturer's protocols for the Purification of Total DNA from nucleated red blood cells and from saliva. To process mosquito abdomens and thoraxes/legs/heads, we modified the kit protocol to include a bead-beating step (2000 strokes/min for 2 min using a GenoGrinder Mini tissue homogenizer) prior to overnight lysis. We quantified the concentration of genomic DNA with a Qubit fluorometer (Invitrogen) and normalized samples to a starting concentration of 2 ng/μL. We quantified *P. relictum* in blood and mosquito samples using a qPCR assay with a hydrolysis probe that targets the cytochrome b gene, optimized for the only lineage of *P. relictum* found in Hawai'i, GRW4[71]. We tested each sample in duplicate or triplicate, with a 40-cycle cut-off, and averaged the cycle threshold (Ct) scores for all detections. A qPCR with a 40 Ct score corresponds to a parasitemia of approximately $1 \times 10^{-7}$ infected RBCs[39].

## Statistical analyses

As noted above, we used the fraction of blood-fed mosquitoes with disseminated infections (any quantity of parasite DNA in the thorax, head, and legs combined) as our measure of host infectiousness because it is strongly correlated with the presence of the vertebrate-infecting malaria life stage[26,72] (sporozoites) in mosquitoes' salivary glands and thus is a strong indicator of a mosquito's ability to transmit malaria. Essentially, we are assuming that if the head/thorax/legs have detectable *Plasmodium* DNA, the mosquito would likely transmit sporozoites if it fed on a host. We used a generalized linear model with a binomial distribution and a logit link to analyze the effect of mosquito age, study year (a categorical predictor that accounts for differences in mosquito populations used), parasitemia, temperature, and days post-feeding and two-way interactions of parasitemia, temperature, and days post-feeding on the fraction of mosquitoes with disseminated infections. We estimated parasitemia (the fraction of red blood cells infected) by converting qPCR Ct value to parasitemia values using a previously published relationship (Logit(Parasitemia) = 17.78 − 0.85*qPCR Ct score[39]). We then tested whether native and introduced bird species differed in parasitemia and in infectiousness using a generalized linear mixed effects model with a beta distribution and species as a random effect using the glmmTMB package[73]. For all statistical tests, we used two-tailed *P*-values.

We assessed the relative importance of the early, high parasitemia and short acute stage of infection in producing infectious mosquitoes compared to the later, lower parasitemia and long chronic stage. We focused on Hawai'i's two most abundant native honeycreepers[37,49,74,75], the Hawai'i 'Amakihi and 'Apapane, for which we had robust sample sizes of parasitemias from wild-caught birds and published experimental infection studies[29,35,57,58,60]. Using the fitted relationship between parasitemia and disseminated infection prevalence just described, we calculated the relative number of mosquitoes that would have disseminated infections 10 days after feeding on birds with high or low phase parasitemias, assuming mosquitoes were held at 24 °C. Ten days is the approximate average lifespan of *Cx. quinquefasciatus* and *Cx. pipiens* mosquitoes in the field[76,77], and 24 °C is an average daily temperature at low elevations in Hawai'i[66]. We used a parasitemia threshold of 1% to separate high and low parasitemias[39].

For the high parasitemia stage, we calculated the average infectiousness of parasitemia values measured throughout its duration, which lasts 38 d in Hawai'i 'Amakihi, and 60 days in 'Apapane[29,35,57,58,60]. For the low parasitemia stage, we averaged infectiousness estimates from wild-caught individuals with parasitemia below 1%. To establish the duration of the low parasitemia phase, we used lifespan data from infected Hawai'i 'Amakihi[78], which, assuming constant type II survival as adults, is 365/(1 − adult annual survival, 0.62) or 961 days, minus the number of days in the high parasitemia stage. Hawai'i 'Amakihi are frequently infected in their first year, have a lower likelihood of infection as adults, and then remain chronically infected for life[36,37,63]. Capture data suggest 'Apapane infections are similar; their adult lifespan and estimated low parasitemia phase were 365/(1 − 0.54) = 789 days[75].

We had no direct information on mosquito feeding patterns, so we estimated the relative host utilization of mosquitoes for different bird species using data on infection prevalence (fraction of infected

individuals). Relative host utilization (sometimes called "feeding preference" or "forage ratio"; we use "host utilization" throughout) refers to whether hosts are fed on in proportion to their abundance (a utilization value of 1), are overutilized (a value > 1) or underutilized (a value < 1), which is influenced by several factors including vector preferences, host defense, and host location[4,79,80]. We corrected prevalence estimates for Hawai'i 'Amakihi and 'Apapane for mortality following infection using a previously developed approach[23]. We first used a generalized linear model with a binomial distribution and a logit link with the infection prevalence of each bird species as the response and site, species, and age as predictors. We used the exponent of the species coefficient (the odds ratio) from this model to infer the relative probability of being fed on because this is the same as the coefficient that would be obtained in a multinomial analysis of feeding patterns where the data points were blood-fed mosquitoes that had fed on that host. Using relative patterns of infection prevalence to estimate mosquito feeding patterns assumes that all species are equally susceptible to infection and, once infected, all species have the same probability of infection being detected by qPCR, which can detect parasitemias above $1 \times 10^{-7}$ parasites/red blood cell[39] (see below for additional discussion of these assumptions). At each site, we calculated the expected fraction of feedings at a site from each species j by multiplying the odds ratio by the relative abundance and dividing it by the sum of this product for all species at a site (feeding fraction = abund$_j$*exp(coef spp$_j$)/$\sum$[abund$_j$*exp(coef spp$_j$)]).

We estimated the infectiousness of different bird communities and the contribution of each species $j$ to transmission using an expression for $R_0$ for a vector borne-pathogen with a single vector[81]:

$$R_{0\,n\,hosts} = \sqrt{\frac{M}{N}\sum_{j=1}^{n}\frac{\beta^2 b s_j c_j \frac{a_j^2}{n_j}\frac{q}{q+\mu_m}}{\mu_m(\mu_j+\gamma_j+\alpha_j)}}$$

with mosquito terms: density ($M$), biting rate ($\beta$), vector competence ($b$), feeding fraction on host j ($a_j$), mortality rate ($\mu_m$); host terms: total density ($N$), susceptibility to infection ($s_j$), average infectiousness ($c_j$), relative abundance ($n_j$), mortality rate ($\mu_j$), recovery rate ($\gamma_j$), disease-caused death rate ($\alpha_j$); and pathogen term: extrinsic incubation period ($1/q$). The *relative* contribution of each bird species to $R_0$ is:

$$Rel.\,Contribution\,to\,R_0 = \frac{a_j^2}{n_j}\frac{s_j c_j}{(\mu_j+\gamma_j+\alpha_j)} = f_j^2 n_j s_j c_j d_j$$

Where $f_j$ is the vector host utilization or forage ratio ($a_j/n_j$) described above (in which host utilization values are inferred from relative infection prevalence), and $d_j$ is the average duration of infectiousness (the inverse of sum of the death and recovery rates). We note that prevalence estimates were only used once in the calculation of species contribution to transmission, to infer $f_j$. We assumed that all species were equally susceptible, $s_j$, based on previous experimental infection studies[29,34,35,56–59], and had similar durations of infectiousness $d_j$. We used our estimates of host infectiousness values described above to estimate $c_j$. These estimates integrate host parasitemias and the estimated probability of mosquitoes having disseminated infections 10 days later from the mosquito experimental infections (Fig. 2). For the 6 species without estimates of parasitemia (most were rare), we used estimates from the nearest taxonomic group (genus, family, order, or class, after averaging empirical data at the same taxonomic level to generate a mean).

### Reporting summary

Further information on research design is available in the Nature Portfolio Reporting Summary linked to this article.

## Data availability

The datasets analyzed in this study can be downloaded from Dryad at https://doi.org/10.5061/dryad.g1jwstr24.

## Code availability

The source code is available on Dryad at https://doi.org/10.5061/dryad.g1jwstr24.

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

## Acknowledgements

We thank the field teams that collected samples over the years, including the U.S. Geological Survey's Avian Malaria Genomic Research Project team and the Hawai'i Vertebrate Introductions and Novel Ecosystems project. We thank the State of Hawai'i Division of Forestry and Wildlife and the Natural Area Reserve System for land access and logistical support. Funding was provided by the National Science Foundation (NSF) grants DEB-1717498 to authors A.M.K., J.T.F., and E.H.P. and DEB 1911853 to authors A.M.K and J.T.F., and a NSF Graduate Research Fellowship to C.M.S. Portions of this article were developed from the thesis of C.M.S. Any use of trade, firm, or product names is for descriptive purposes only and does not imply endorsement by the U.S. Government.

## Author contributions

C.M.S. and A.M.K.: conceptualization, data curation, formal analysis, investigation, methodology, validation, visualization, funding acquisition, writing—original draft, review, editing; K.L.P. and K.B.: data curation, investigation; I.J.I., S.L., D.H., R.M., F.C.F., E.C.A.: investigation; J.T.F. and E.H.P.: funding acquisition, resources, writing—review and editing; All authors gave final approval for publication and agreed to be held accountable for the work performed therein.

## Competing interests

The authors declare no competing interests.
