## [Transparent Peer Review file · Nature Communications]

Variation in pathogen load and the pathogen load–infectiousness relationship broaden avian malaria’s distribution

Corresponding Author: Dr Christa Seidl

Version 0:

Reviewer comments:

Reviewer #1

(Remarks to the Author)

This manuscript investigates the role of within-species variation in pathogen load and the shape of the pathogen load–infectiousness relationship in influencing avian malaria transmission across Hawaiian bird communities. The authors use a combination of experimental infections in canaries, parasitemia data from wild bird species, estimates of mosquito feeding preferences (inferred from parasite prevalence), and community composition data to evaluate bird species contributions to malaria transmission.

They report that the relatively gradual slope of the parasitemia–infectiousness relationship, together with substantial within-species variability in parasitemia, lead to broad overlaps of infectiousness across both native and introduced bird species. They also highlight the potential importance of chronic low-parasitemia infections and mosquito feeding preferences (as inferred) in driving community-level transmission.

I believe the study addresses an important ecological and conservation question on the persistence and spread of avian malaria in Hawai‘i. It raises a novel idea on how within host species variation of pathogen load interact with vector susceptibility to infection, and later dissemination (shape of pathogen load – infectiousness; Figure 1 in manuscript). And it provides a novel set of transmission trait data. The manuscript is well written and organized. I also like that the authors also discuss the results in a broader comparative context with other human pathogens (e.g., West Nile virus, dengue virus).

There are some limitations (including a few that has already been pointed out by the authors) namely:

- The Methods state that *Culex quinquefasciatus* egg rafts were collected from three locations and that wild-collected, lab-reared adults of various ages were used in experimental feedings. However, it is not clear which mosquito populations and ages were used in each bird feeding (and parasitemia–temperature–day combination). Is there potential for inter-population and age-dependent differences in vector susceptibility? Are they assuming all mosquitoes are equally susceptible to infection? Could they clarify the mapping of mosquito population and age to experimental trials and indicate whether these factors were balanced, randomized, or modeled? The primary GLM appears to include parasitemia, temperature, days since feeding, and the two-way interactions (no random effects for mosquito population or age).
- The approach of inferring mosquito feeding preferences from relative infection prevalence is creative, but it is also heavily assumption-dependent. Specifically, it assumes equal susceptibility, equal infection duration, and equal detectability across species, so that prevalence differences reflect only differences in mosquito feeding rates. In practice, prevalence also depends on survival after infection, immune clearance, host age structure, and perhaps other factors. I believe authors tried to address many of these points already by including them in the models/data analysis. But I recommend clarifying these limitations more explicitly. It may also help to frame the metric as “inferred feeding preference” rather than just “feeding preference” (e.g., in figure axis labels) to emphasize that this is a proxy rather than a direct measure from blood meal data.
- The framework combines species infection prevalence (used to infer mosquito feeding preferences) with parasitemia distributions to estimate species’ contributions to transmission. I am slightly concerned that this may introduce some degree of circularity. If prevalence is already partly determined by mosquito feeding rates, and then prevalence (via “inferred feeding preference”) is used again to weight transmission estimates, this may amplify the influence of prevalence. In other word,

because prevalence already reflects (at least partly) mosquito feeding rates, there is some sort of circularity: prevalence influences the outcome both directly (as observed infection) and indirectly (as inferred feeding preference). I would love to hear the authors' thoughts on this or clarifying how this is accounted for, or demonstrating that prevalence is not effectively "counted twice". The authors should also discuss how real feeding data (e.g., blood meal analysis) might provide independent validation.

- The authors aim to quantify bird "infectiousness" as a function of parasitemia. I recommend defining "infectiousness" very clearly early in the manuscript. Is it the bird's ability to transmit parasites to other birds (via mosquito vectors), or its ability to transmit parasites specifically to mosquito vectors? The manuscript appears to adopt the former definition, since the outcome of interest is disseminated infection in mosquitoes. However, by using dissemination as the endpoint, the measure of infectiousness also incorporates vector traits (e.g., midgut escape) that may not be directly related to host parasitemia. It would strengthen the manuscript to justify this choice explicitly (or by giving clear definition of infectiousness) and perhaps also present midgut infection results in the main manuscript in parallel as a more direct measure of host-to-vector transfer.
- The slope of the parasitemia–infectiousness relationship is described as relatively gradual (Line 146). I wonder if this pattern partly reflects averaging across highly variable individual-level outcomes. In other words, heterogeneity in the underlying parasitemia–infectiousness relationships may be obscured when data are pooled, producing an apparently gradual curve. If so, the interpretation of "no clear threshold" may need to be qualified: the gradual slope could be statistical rather than biological. It would help to discuss how such heterogeneity might affect conclusions about transmission dynamics in natural systems.

Other comments

In the abstract, I suggest change some wordings to be clear about the methods to avoid implying direct measurements when values were inferred

- Somewhere between Line 34-36, the authors should specify that canaries were used as the model organism for experimental infections.
- Line 37, I suggest changing "estimated the infectiousness..." to "inferred the infectiousness..."
- Line 38, please clarify here that mosquito feeding preferences were inferred from infection prevalence.

When referring to the results of binomial models (first section of Result), the phrasing "fraction of mosquitoes with disseminated infection" is not appropriate. E.g., in the fitted lines & intervals in Fig 2 should be referred as probability. The raw data can be fractions/proportions, but the modeled outcome is probability.

In Figure 2, the x-axis is labeled "parasitemia, % cells infected," but the values plotted (1e-6 to 1e-0). Please clarify if this label should be percentage or proportion.

Line 132-133: a bit pedantic here but the way the sentence is currently written makes it sound as though small sample sizes caused the variability. In reality, the variability likely reflects inherent biological variation, while the small sample sizes increased the noise in the estimates.

Line 145-148: confusing phrasing. Maybe something like "increasing parasitemia across a 10⁵-fold range (from 10⁻⁶ to 10⁻¹) increased mosquito disseminated infection 10 days after feeding by only ~3.6-fold, from X% to X%..."

For the model method to create Figure 5, please clarify the parameter values that were used for competence and abundance. To help readers understand why setting competence and abundance as 1s doesn't change the model results. Also, in Figure 5, the y-axis is labeled "contribution to transmission." Since these are model-derived and normalized values, I recommend labeling as "Estimated relative contribution to transmission" (or similar) to avoid implying that these are directly observed measures.

I also recommend using the term "inferred feeding preference" as much as appropriate. This would be especially important in prominent places such as figure axis labels (e.g., Figure 4) and key results statements.

Similarly, in Figure 4 and Figure 3B, label "infectiousness" may cause a reader to misinterpret that these were directly measured. Since these values are inferred from the canary experimental model applied to field parasitemia data, a term such as "inferred infectiousness" might be more accurate.

Lines 278–279: The statement that "a steeper threshold and lower variability in parasitemia ... would have resulted in a far more limited geographic distribution and host range" makes intuitive sense, but the mechanism could be explained more explicitly. The idea seems to be that a sharp threshold plus low variability would confine transmission to a narrow subset of high-parasitemia species, whereas a gradual slope and high variability allow many species to contribute at least occasionally. Clarifying this reasoning at this point in the Discussion would strengthen the reader's understanding.

Could authors discussed a bit how if different parasite strains/lineage are circulating in either in Hawai'i or elsewhere, and how if such differences exist but are ignored, that could introduce extra variability or bias in the inferred infectiousness–parasitemia relationship. For clarity, I also suggest explicitly stating early in the Introduction (e.g., around Line 99-100) that GRW4 is the most common lineage in Hawaiian birds with limited genetic variation. This would help readers avoid assuming the results apply to all settings.

Reviewer #2

(Remarks to the Author)

Overall comments

This is a massive study which took a lot of effort to pull off. It helps to resolve some mysteries that have persisted in Hawaii regarding avian malaria and the threat to native avifauna. I enjoy these kinds of studies that integrate field and lab data to infer community disease dynamics, such as which bird species or which bird communities are most important for avian malaria transmission. The authors have done a good job explaining these analyses and acknowledging assumptions and limitations. Below I have outlined a few areas of concern for the authors to consider.

The authors repeatedly use the term 'preference' throughout this entire manuscript when referring to mosquito feeding patterns on different birds but it appears that these are an inappropriate use of the term. My specific comments below provide a review paper to help the authors select the best term to use that would be more appropriate. For example, the authors are suggesting that house finches are the 'most preferred' hosts given that they had the highest infection prevalence and susceptibility is similar among species. However, mosquito feeding patterns on house sparrows could be higher for a variety of extrinsic factors such as ease of access (where they roost and at what height) as well as the lack of anti-mosquito behavior that other bird species might have. These extrinsic factors have nothing to do with 'preference'.

Throughout the manuscript, the authors are referring to '% disseminated' as 'infectiousness'. However, from that I can tell in the experimental methods, it looks like infected mosquitoes had the abdomen removed from the head/thorax/legs and then both sample types were tested for Plasmodium using a qPCR. While the authors converted the qPCR ct value to parasitemia for the birds, I don't see any use of the actual ct-value for the mosquito abdomens or rest of the body. I assume the persistent presence of Plasmodium DNA in the abdomen many days post-exposure imply sporogony is happening in the mosquito. Then the presence of DNA in the head/thorax could imply oocysts or sporozoites could be present but it almost appears like the authors are simply judging the head/thorax as either being DNA pos or neg for Plasmodium. The number of oocysts and sporozoites can vary tremendously in the Culex vector so it would have been nice if dissections were done to quantify these different life stages and numbers to come up with a better measure of 'infectiousness'. I think the authors need to be cautious with their use of terms and some of the interpretations given this limitation.

One additional caveat with this study is that the authors are making several simplifying assumptions that could be misleading. For example, they are assuming that bird parasitemia directly relates to the probability of mosquitoes becoming infected. However, a recent study, which the authors cite in the discussion, found that different bird species do not have the same relationship of viremia to the probability of Culex mosquito infection with West Nile virus. For whatever reason, American robins can infect more Culex with the same viremia as common grackles. The laboratory experiments in this current study were all done with canaries and the authors also used some experimental infection data from prior challenges of other bird species. The authors should discuss the possibility that parasitemias in these birds are not all 'equal' with respect to the importance of avian malaria transmission.

Specific comments

Ln. 42. 'created more infectious mosquitoes' sounds like it would be more appropriate to say 'resulted in more infectious mosquitoes'.

Ln. 43. Is 'feeding preference' the correct term.....implies host choice experiments.

Ln. 63. It is not just host preference, an intrinsic factor, but also extrinsic factors such as host availability that determine feeding patterns in nature. Reading the following study would help ensure you are using the correct term in the correct context:

Fikrig K, Harrington LC. Understanding and interpreting mosquito blood feeding studies: the case of *Aedes albopictus*. *Trends Parasitol.* 2021 Nov;37(11):959-975. doi: 10.1016/j.pt.2021.07.013.

Ln. 64. In this context, I would think saying 'vector feeding patterns' would be more appropriate than 'vector feeding preference'. You are referring to what happens in nature and vector feeding patterns in nature are influenced by preference as well as additional factors.

Ln. 499-502. It is good that you say you did not have mosquito feeding pattern data since you did not collect Culex in Hawaii and conduct the bloodmeal analysis. But this is where you introduce the concept of 'feeding preference' but you don't explain where this is coming from (reference 53 is a bird infection study). Maybe what you mean in this context would be non-random mosquito feeding patterns such that Cx. quinq. utilize some birds in equal proportion to availability, some birds are overutilized, and others are underutilized. I assume this is what you are referring to in this context?

Ln. 508-511. I don't follow this concept of using plasmodium infection of the birds to estimate mosquito feeding patterns. This sentence cites the Seidl et al. 2024 study but I am not seeing anything in that past study linking bird infection to mosquito feeding patterns. Therefore, the 'feeding fraction' equation on Ln 515 seems to mainly just assume mosquitoes are feeding in equal proportion to bird availability at a given site. Is this the case?

Figure S11. I don't understand this figure. How do you have the contribution of bird species to transmission using feeding preferences measured directly? That would imply some kind of host choice study allowing mosquitoes to select different bird

species which the authors didn't do. I don't actually know what each dot represents; are these different bird species?

Reviewer #3

(Remarks to the Author)

"Intraspecific variation in pathogen load and gradual pathogen load-infectiousness relationship shape broad distribution of avian malaria in Hawai'i". Seidl et al

Summary: This is a well written description of an immense amount of field and laboratory data. The effort involved is tremendous. The ability to take empirical data generated from a variety of experiments and observations and pull them together to examine the community level transmission and the role of each species is very original. However, the results do not lead to a very clear conclusion and are not sufficiently put into a more general context. Much of the discussion is narrowly focused on the specifics of the system in Hawai'i. This leads the reader to focus on the deficiencies in the data (well articulated by the authors) instead of how the results are really important even considering the challenges. As noted, this is an incredible dataset representing a huge amount of work, so it is a shame that the conclusions do not seem broadly important.

Specific Comments: As noted, this is a well-written manuscript, so I had very few grammatical/fluency, etc. suggestions.

Title: The title was not easy to parse on its own. "Gradual" could have several interpretations here (over time? Over another gradient? What is gradual (a shallow slope, but how shallow is gradual?). Likewise, is "broad" meant to mean everywhere in Hawai'i? Or is this a type of distribution? The title speaks to the lack of a clear results—not the fault of the authors, sometimes the data are just like that, but it doesn't make this attractive to read.

Abstract: Fine. One small comment is to be sure when "species" is used, the authors should make sure they mean the bird hosts (not the vector, which also has a pathogen load, etc.).

Introduction: Again, the writing is very clear. However, the first paragraph essentially quickly pivots from a fundamental question in biology (specialization of parasites) to a methodological one "These data can be difficult to obtain..." And then continues to delve into these challenges. In that is lost the more interesting motivating questions and reasons for the work. The second paragraph (and figure 1) does bring this back around to something of somewhat general interest, but I think an opportunity was missed here by not providing some analogies to other pathogen-vector-host systems. West Nile is mentioned in the discussion, but not in the introduction, and this seems like a missed opportunity. Likewise, there may be analogies to other wildlife diseases or even human pathogens (although obviously human pathogens only deal with one species of host, there are many reasons there may be variation in pathogen load in different people, and a vast literature on this).

Results: The amount of data is astounding. The results are generally very good, although not all of the figures were very revealing, except to demonstrate how much effort went into this. As repeatedly admitted by the authors, the feeding preference results seem to be a stretch, almost teleological, which does give one pause (and a little hard to parse how this is impacted as a composite variable with regards to the estimation of contribution to R0). That being said, pulling together all of these factors to make these estimations is an exciting result. Side note: I found figure 5 hard to read in my copy. The subtlety in color gradation was a challenge. I suppose the point was just to show the variation between sites in what species are contributing—an important, but ultimately unsatisfying result.

Discussion. I felt the structure of the discussion was not optimal. The first sentence is confusing as written (63 of 64 sites is unclear—was the 64th one not sampled well?). The "despite" would seem to reflect would seem to suggest there was a priori prediction that variable (or diverse?) host communities should somehow result in less transmission variability? Anyway, this set the tone for the discussion.

One wonders a little about either focusing more on what might explain variation in host contribution besides native versus not native. Size? Behavior? Or is it idiosyncratic? This wasn't really mentioned, which seems an oversight.

The idea of long-term, low intensity infections is built into many transmission systems. Oft ignored, vector-borne filaria, in large part, adopt this strategy, as do most species of human malaria (the authors reference *P. falciparum*, but it is the other species of malaria, especially *P. vivax* that are the master's of hiding out in a host, right?).

The part of the discussion that focuses on the specifics of the individual species (house finch—side note, why is this capitalized? Unless it is named after a Dr. House, I think house finch is fine) drifts into areas that are not of general interest.

Methods. No comments, but I do note the tremendous amount of work that went into this study.

Version 1:

Reviewer comments:

Reviewer #1

(Remarks to the Author)

Thank you for the responses. I've reviewed the authors point-by-point reply and I appreciate that they have carefully addressed the comments.

Overall, I find that the authors have responded comprehensively including the re-analysis incorporating mosquito age and population, clarified their assumptions, and expanded explanations as suggested. They also made the recommended wording and figure label adjustments.

I believe these changes have substantially strengthened the manuscript's clarity. At this stage, the revised version satisfactorily addresses all my comments and I have no further substantive suggestions.

Reviewer #2

(Remarks to the Author)

The first round of reviews was comprehensive and the author's responses to the comments were excellent. I read over the revised manuscript and the clarifying details and other changes help to improve the manuscript. I have no further comments.

Reviewer #3

(Remarks to the Author)

Kudos. You have done a good job addressing my concerns, to the best the data can allow.

One suggestion for a title:

Variation in host pathogen load and its relationship to infectiousness result in a broad distribution of infections: insights from avian malaria.

I am not sure it is much better, but I hope it inspires some noodling around by the authors.

Yes, that one figure is more garish, but also much more readable (to my eyes, anyway).

REVIEWER COMMENTS

Reviewer #1 (Remarks to the Author):

This manuscript investigates the role of within-species variation in pathogen load and the shape of the pathogen load–infectiousness relationship in influencing avian malaria transmission across Hawaiian bird communities. The authors use a combination of experimental infections in canaries, parasitemia data from wild bird species, estimates of mosquito feeding preferences (inferred from parasite prevalence), and community composition data to evaluate bird species contributions to malaria transmission.

They report that the relatively gradual slope of the parasitemia–infectiousness relationship, together with substantial within-species variability in parasitemia, lead to broad overlaps of infectiousness across both native and introduced bird species. They also highlight the potential importance of chronic low-parasitemia infections and mosquito feeding preferences (as inferred) in driving community-level transmission.

I believe the study addresses an important ecological and conservation question on the persistence and spread of avian malaria in Hawai'i. It raises a novel idea on how within host species variation of pathogen load interact with vector susceptibility to infection, and later dissemination (shape of pathogen load – infectiousness; Figure 1 in manuscript). And it provides a novel set of transmission trait data. The manuscript is well written and organized. I also like that the authors also discuss the results in a broader comparative context with other human pathogens (e.g., West Nile virus, dengue virus).

Response: We appreciate the supportive comments.

There are some limitations (including a few that has already been pointed out by the authors) namely:

- The Methods state that *Culex quinquefasciatus* egg rafts were collected from three locations and that wild-collected, lab-reared adults of various ages were used in experimental feedings. However, it is not clear which mosquito populations and ages were used in each bird feeding

(and parasitemia–temperature–day combination). Is there potential for inter-population and age-dependent differences in vector susceptibility? Are they assuming all mosquitoes are equally susceptible to infection? Could they clarify the mapping of mosquito population and age to experimental trials and indicate whether these factors were balanced, randomized, or modeled? The primary GLM appears to include parasitemia, temperature, days since feeding, and the two-way interactions (no random effects for mosquito population or age).

Response: Thank you for raising these issues. We re-analyzed the mosquito infection data while including both mosquito age and population. The analyses were qualitatively identical to those originally presented. There were some minor differences among mosquito populations and a positive effect of age on susceptibility, but the significance of all other predictors was identical, with only minor differences in coefficients, all of which are updated in the manuscript and supplemental material. We've also updated our online Dryad code file to reflect this revised analysis.

- The approach of inferring mosquito feeding preferences from relative infection prevalence is creative, but it is also heavily assumption-dependent. Specifically, it assumes equal susceptibility, equal infection duration, and equal detectability across species, so that prevalence differences reflect only differences in mosquito feeding rates. In practice, prevalence also depends on survival after infection, immune clearance, host age structure, and perhaps other factors. I believe authors tried to address many of these points already by including them in the models/data analysis. But I recommend clarifying these limitations more explicitly. It may also help to frame the metric as “inferred feeding preference” rather than just “feeding preference” (e.g., in figure axis labels) to emphasize that this is a proxy rather than a direct measure from blood meal data.

Response: We agree completely and have revised the figure axis label in Fig 4 as suggested.

- The framework combines species infection prevalence (used to infer mosquito feeding preferences) with parasitemia distributions to estimate species' contributions to transmission. I am slightly concerned that this may introduce some degree of circularity. If prevalence is already partly determined by mosquito feeding rates, and then prevalence (via “inferred feeding preference”) is used again to weight transmission estimates, this may amplify the influence of prevalence. In other words, because prevalence already reflects (at least partly) mosquito feeding rates, there is some sort of circularity: prevalence influences the outcome both directly (as observed infection) and indirectly (as inferred feeding preference). I would love to hear the authors' thoughts on this or clarifying how this is accounted for, or demonstrating that prevalence is not effectively “counted twice”. The authors should also discuss how real feeding data (e.g., blood meal analysis) might provide independent validation.

Response: We appreciate the reviewer raising this issue. Prevalence is only used once, in the calculation of species' contributions to transmission (in the inferred feeding preferences, f_j in the Relative contributions to R_0 equation in the Methods). The other components of the calculation are the relative abundance of the species and the infectiousness (based on measured parasitemia values). We have added text clarifying this issue to the Methods since other readers may wonder the same thing.

• The authors aim to quantify bird “infectiousness” as a function of parasitemia. I recommend defining “infectiousness” very clearly early in the manuscript. Is it the bird’s ability to transmit parasites to other birds (via mosquito vectors), or its ability to transmit parasites specifically to mosquito vectors? The manuscript appears to adopt the former definition, since the outcome of interest is disseminated infection in mosquitoes. However, by using dissemination as the endpoint, the measure of infectiousness also incorporates vector traits (e.g., midgut escape) that may not be directly related to host parasitemia. It would strengthen the manuscript to justify this choice explicitly (or by giving clear definition of infectiousness) and perhaps also present midgut infection results in the main manuscript in parallel as a more direct measure of host-to-vector transfer.

Response: We agree with the reviewer that this is an important and potentially confusing issue! We have added a clear definition of what we mean by infectiousness. We note that it does incorporate vector traits (e.g., midgut escape). The reason for this is that the extent of midgut escape depends on host parasitemia. As a result, if we only defined infectiousness using the midgut infections, the importance of parasitemia (and differences among hosts) would be substantially underestimated.

• The slope of the parasitemia–infectiousness relationship is described as relatively gradual (Line 146). I wonder if this pattern partly reflects averaging across highly variable individual-level outcomes. In other words, heterogeneity in the underlying parasitemia–infectiousness relationships may be obscured when data are pooled, producing an apparently gradual curve. If so, the interpretation of “no clear threshold” may need to be qualified: the gradual slope could be statistical rather than biological. It would help to discuss how such heterogeneity might affect conclusions about transmission dynamics in natural systems.

Response: The reviewer is correct that if there is substantial heterogeneity among mosquitoes in susceptibility and midgut escape, this will flatten the relationship between parasitemia and the fraction of mosquitoes with disseminated infections (our measure of “infectiousness” as noted above). We have added text reflecting this point. However, as we also note in the text, heterogeneity in susceptibility is a biological effect not a statistical one, and the extent of heterogeneity can vary across a wide or narrow range of host parasitemias. As we previously described in the first paragraph of the Discussion, the slopes of these relationships can vary among mosquito species and pathogens (with some being quite steep), demonstrating that a gradual slope is not a statistical certainty. Thus, the gradual relationship we found is a biological pattern, and one with quite large consequences for transmission. We have added references in the added text to link the reader to the rich theory and some empirical data on this very interesting topic.

Other comments

In the abstract, I suggest change some wordings to be clear about the methods to avoid implying direct measurements when values were inferred

- Somewhere between Line 34-36, the authors should specify that canaries were used as the model organism for experimental infections.

Response: We have revised the text as suggested.

- Line 37, I suggest changing “estimated the infectiousness...” to “inferred the infectiousness...”

Response: We have revised the text as suggested.

- Line 38, please clarify here that mosquito feeding preferences were inferred from infection prevalence.

Response: We have revised the text as suggested.

When referring to the results of binomial models (first section of Result), the phrasing “fraction of mosquitoes with disseminated infection” is not appropriate. E.g., in the fitted lines & intervals in Fig 2 should be referred as probability. The raw data can be fractions/proportions, but the modeled outcome is probability.

Response: We have revised the text in the Results and Fig 2 legend as suggested.

In Figure 2, the x-axis is labeled “parasitemia, % cells infected,” but the values plotted ($1e-6$ to $1e-0$). Please clarify if this label should be percentage or proportion.

Response: We appreciate the reviewer’s careful eye. We have corrected the x-axis.

Line 132-133: a bit pedantic here but the way the sentence is currently written makes it sound as though small sample sizes caused the variability. In reality, the variability likely reflects inherent biological variation, while the small sample sizes increased the noise in the estimates.

Response: We have revised the text as suggested, by replacing “variability” with “observation error”.

Line 145-148: confusing phrasing. Maybe something like “increasing parasitemia across a 10^5 -fold range (from 10^{-6} to 10^{-1}) increased mosquito disseminated infection 10 days after feeding by only ~3.6-fold, from X% to X%...”

Response: We have revised the text as suggested.

For the model method to create Figure 5, please clarify the parameter values that were used for competence and abundance. To help readers understand why setting competence and abundance as 1s doesn’t change the model results.

Response: We have added text to the supplemental material as suggested to clarify the simulations comparing methods for estimating mosquito host preferences on the role of species in transmission. Competence and abundance definitely affect host contributions to transmission, but are independent of the methods used to estimate mosquito feeding preferences, and we measured both of these in our study.

Also, in Figure 5, the y-axis is labeled “contribution to transmission.” Since these are model-derived and normalized values, I recommend labeling as “Estimated relative contribution to transmission” (or similar) to avoid implying that these are directly observed measures.

Response: We have revised the text here and elsewhere as suggested (both Figure Y-axes, as well as Results text).

I also recommend using the term “inferred feeding preference” as much as appropriate. This would be especially important in prominent places such as figure axis labels (e.g., Figure 4) and key results statements.

Response: We have revised the text here and elsewhere as suggested (both Figure Y-axes, as well as Results text).

Similarly, in Figure 4 and Figure 3B, label “infectiousness” may cause a reader to misinterpret that these were directly measured. Since these values are inferred from the canary experimental model applied to field parasitemia data, a term such as “inferred infectiousness” might be more accurate.

Response: We have revised the text here and elsewhere as suggested (both Figure Y-axes, as well as Results text).

Lines 278–279: The statement that “a steeper threshold and lower variability in parasitemia ... would have resulted in a far more limited geographic distribution and host range” makes intuitive sense, but the mechanism could be explained more explicitly. The idea seems to be that a sharp threshold plus low variability would confine transmission to a narrow subset of high-parasitemia species, whereas a gradual slope and high variability allow many species to contribute at least occasionally. Clarifying this reasoning at this point in the Discussion would strengthen the reader’s understanding.

Response: We agree and have added text around this statement to explain this more clearly.

Could authors discuss a bit how if different parasite strains/lineage are circulating in either in Hawai’i or elsewhere, and how if such differences exist but are ignored, that could introduce extra variability or bias in the inferred infectiousness–parasitemia relationship. For clarity, I also suggest explicitly stating early in the Introduction (e.g., around Line 99-100) that GRW4 is the most common lineage in Hawaiian birds with limited genetic variation. This would help readers avoid assuming the results apply to all settings.

Response: Yes, we have added text to address both of these issues, as suggested.

Reviewer #2 (Remarks to the Author):

Overall comments

This is a massive study which took a lot of effort to pull off. It helps to resolve some mysteries that have persisted in Hawaii regarding avian malaria and the threat to native avifauna. I enjoy these kinds of studies that integrate field and lab data to infer community disease dynamics, such as which bird species or which bird communities are most important for avian malaria transmission. The authors have done a good job explaining these analyses and acknowledging assumptions and limitations. Below I have outlined a few areas of concern for the authors to consider.

Response: Thank you, we appreciate the supportive comments.

The authors repeatedly use the term 'preference' throughout this entire manuscript when referring to mosquito feeding patterns on different birds but it appears that these are an inappropriate use of the term. My specific comments below provide a review paper to help the authors select the best term to use that would be more appropriate. For example, the authors are suggesting that house finches are the 'most preferred' hosts given that they had the highest infection prevalence and susceptibility is similar among species. However, mosquito feeding patterns on house sparrows could be higher for a variety of extrinsic factors such as ease of access (where they roost and at what height) as well as the lack of anti-mosquito behavior that other bird species might have. These extrinsic factors have nothing to do with 'preference'.

Response: We agree, consulted your suggested review paper (Fikrig & Harrington, *Parasit. Vectors* 2021), and have revised the terminology. We now use "host utilization" rather than "feeding preferences"

Throughout the manuscript, the authors are referring to '% disseminated' as 'infectiousness'. However, from that I can tell in the experimental methods, it looks like infected mosquitoes had the abdomen removed from the head/thorax/legs and then both sample types were tested for Plasmodium using a qPCR. While the authors converted the qPCR ct value to parasitemia for the birds, I don't see any use of the actual ct-value for the mosquito abdomens or rest of the body. I assume the persistent presence of Plasmodium DNA in the abdomen many days post-exposure imply sporogony is happening in the mosquito. Then the presence of DNA in the head/thorax could imply oocysts or sporozoites could be present but it almost appears like the authors are simply judging the head/thorax as either being DNA pos or neg for Plasmodium. The number of oocysts and sporozoites can vary tremendously in the Culex vector so it would have been nice if dissections were done to quantify these different life stages and numbers to come up with a better measure of 'infectiousness'. I think the authors need to be cautious with their use of terms and some of the interpretations given this limitation.

Response: The reviewer is correct that we scored mosquito head/thorax/legs as DNA positive or negative for *Plasmodium relictum*. We have added text to the Methods (see statistical analyses section), as suggested, to clarify this, and stated our assumption more clearly to be fully transparent.

One additional caveat with this study is that the authors are making several simplifying assumptions that could be misleading. For example, they are assuming that bird parasitemia directly relates to the probability of mosquitoes becoming infected. However, a recent study, which the authors cite in the discussion, found that different bird species do not have the same relationship of viremia to the probability of Culex mosquito infection with West Nile virus. For whatever reason, American robins can infect more Culex with the same viremia as common grackles. The laboratory experiments in this current study were all done with canaries and the authors also used some experimental infection data from prior challenges of other bird species. The authors should discuss the possibility that parasitemias in these birds are not all 'equal' with respect to the importance of avian malaria transmission.

Response: We agree! We have highlighted this issue in the Discussion in the paragraph that begins: "Four limitations affected this study."

Specific comments

Ln. 42. 'created more infectious mosquitoes' sounds like it would be more appropriate to say 'resulted in more infectious mosquitoes'.

Response: We have revised the text, as suggested.

Ln. 43. Is 'feeding preference' the correct term.....implies host choice experiments.

Response: We have revised the terminology to "host utilization" throughout the paper.

Ln. 63. It is not just host preference, an intrinsic factor, but also extrinsic factors such as host availability that determine feeding patterns in nature. Reading the following study would help ensure you are using the correct term in the correct context:

Fikrig K, Harrington LC. Understanding and interpreting mosquito blood feeding studies: the case of *Aedes albopictus*. *Trends Parasitol.* 2021 Nov;37(11):959-975. doi: 10.1016/j.pt.2021.07.013.

Response: Thank you for this study. We have revised the terminology throughout the paper, as suggested, and cite this study in the Methods.

Ln. 64. In this context, I would think saying 'vector feeding patterns' would be more appropriate than 'vector feeding preference'. You are referring to what happens in nature and vector feeding patterns in nature are influenced by preference as well as additional factors.

Response: We have revised the terminology to "host utilization" throughout the paper, as suggested.

Ln. 499-502. It is good that you say you did not have mosquito feeding pattern data since you did not collect *Culex* in Hawaii and conduct the bloodmeal analysis. But this is where you introduce the concept of 'feeding preference' but you don't explain where this is coming from (reference 53 is a bird infection study). Maybe what you mean in this context would be non-random mosquito feeding patterns such that *Cx. quinq.* utilize some birds in equal proportion to availability, some birds are overutilized, and others are underutilized. I assume this is what you are referring to in this context?

Response: We have added a sentence here to explain this in more detail, as suggested and have added references including Fikrig & Harrington, *Parasit. Vectors* 2021.

Ln. 508-511. I don't follow this concept of using plasmodium infection of the birds to estimate mosquito feeding patterns. This sentence cites the Seidl et al. 2024 study but I am not seeing anything in that past study linking bird infection to mosquito feeding patterns. Therefore, the 'feeding fraction' equation on Ln 515 seems to mainly just assume mosquitoes are feeding in equal proportion to bird availability at a given site. Is this the case?

Response: No, we do not assume mosquitoes feed on hosts in proportion to bird availability. In this paragraph, we describe how we used relative patterns of infection prevalence to estimate host utilization. The patterns of infection prevalence are shown in Figure S5. The estimated host utilization values are calculated using the exponent of the coefficient from the model fit to the infection prevalence data, which the paragraph explains. We've improved some of the wording so this is more clear. The host utilization

values are shown in Figure 3C and are not = 1 (which would reflect feeding in proportion to host availability).

Figure S11. I don't understand this figure. How do you have the contribution of bird species to transmission using feeding preferences measured directly? That would imply some kind of host choice study allowing mosquitoes to select different bird species which the authors didn't do. I don't actually know what each dot represents; are these different bird species?

We have added text in both the Supplemental Methods and the figure legend to clarify what this figure shows. It is a simulation, comparing the accuracy of calculations of host contributions to transmission using two methods for estimating mosquito host utilization: 1) using relative infection prevalence (as we did in this study), and 2) the more frequently used method for measuring mosquito relative host utilization - identifying the source of engorged/blood-fed mosquitoes. Figure 11 shows that both methods are equally accurate for estimating host contributions to transmission, because using relative infection prevalence to estimate mosquito relative host utilization and using the resulting estimates to estimate host contributions to transmission accounts (unmeasured) variation in susceptibility and the duration of infection.

Reviewer #3 (Remarks to the Author):

"Intraspecific variation in pathogen load and gradual pathogen load--infectiousness relationship shape broad distribution of avian malaria in Hawai'i". Seidl et al

Summary: This is a well written description of an immense amount of field and laboratory data. The effort involved is tremendous. The ability to take empirical data generated from a variety of experiments and observations and pull them together to examine the community level transmission and the role of each species is very original. However, the results do not lead to a very clear conclusion and are not sufficiently put into a more general context. Much of the discussion is narrowly focused on the specifics of the system in Hawai'i. This leads the reader to focus on the deficiencies in the data (well articulated by the authors) instead of how the results are really important even considering the challenges. As noted, this is an incredible dataset representing a huge amount of work, so it is a shame that the conclusions do not seem broadly important.

Response: Thank you. We have revised the discussion substantially to broaden the focus on the conceptual issues that are broader than the local system. We also look into work to other systems, including West Nile virus, dengue, human malaria.

Specific Comments: As noted, this is a well-written manuscript, so I had very few grammatical/fluency, etc. suggestions.

Response: Thank you.

Title: The title was not easy to parse on its own. "Gradual" could have several interpretations here (over time? Over another gradient? What is gradual (a shallow slope, but how shallow is

gradual?). Likewise, is “broad” meant to mean everywhere in Hawai’i? Or is this a type of distribution? The title speaks to the lack of a clear results—not the fault of the authors, sometimes the data are just like that, but it doesn’t make this attractive to read.

Response: We appreciate the reviewer’s comments. We have revised the title, keeping within the word limit, and if the reviewer has suggestions for improving the title we’d welcome them!

Abstract: Fine. One small comment is to be sure when “species” is used, the authors should make sure they mean the bird hosts (not the vector, which also has a pathogen load, etc.).

Response: We have revised the text to clarify this, as suggested.

Introduction: Again, the writing is very clear. However, the first paragraph essentially quickly pivots from a fundamental question in biology (specialization of parasites) to a methodological one “These data can be difficult to obtain...” And then continues to delve into these challenges. In that is lost the more interesting motivating questions and reasons for the work. The second paragraph (and figure 1) does bring this back around to something of somewhat general interest, but I think an opportunity was missed here by not providing some analogies to other pathogen-vector-host systems. West Nile is mentioned in the discussion, but not in the introduction, and this seems like a missed opportunity. Likewise, there may be analogies to other wildlife diseases or even human pathogens (although obviously human pathogens only deal with one species of host, there are many reasons there may be variation in pathogen load in different people, and a vast literature on this).

Response: We have added text, as suggested, to broaden the work to other systems, including West Nile virus, dengue, human malaria, and others.

Results: The amount of data is astounding. The results are generally very good, although not all of the figures were very revealing, except to demonstrate how much effort went into this. As repeatedly admitted by the authors, the feeding preference results seem to be a stretch, almost teleological, which does give one pause (and a little hard to parse how this is impacted as a composite variable with regards to the estimation of contribution to R_0). That being said, pulling together all of these factors to make these estimations is an exciting result.

Response: We appreciate the positive comments.

Side note: I found figure 5 hard to read in my copy. The subtlety in color gradation was a challenge. I suppose the point was just to show the variation between sites in what species are contributing—an important, but ultimately unsatisfying result.

Response: We have revised this figure to use a more distinct color palette, as suggested. The aesthetics are not beautiful, but the interpretability is improved. We have also added a note to the legend to direct the reader to Figure 4D, which shows the individual species contributions in a simpler and more comparable way.

Discussion. I felt the structure of the discussion was not optimal. The first sentence is confusing as written (63 of 64 sites is unclear—was the 64th one not sampled well?). The “despite” would seem to reflect would seem to suggest there was a priori prediction that variable (or diverse?)

host communities should somehow result in less transmission variability? Anyway, this set the tone for the discussion.

Response: Thank you, we see the confusion. We have revised the text to improve the readability and make our points clearer. We were suggesting that highly variable host communities would be expected to result in only a subset of sites having transmission. In contrast, we found avian malaria essentially everywhere.

One wonders a little about either focusing more on what might explain variation in host contribution besides native versus not native. Size? Behavior? Or is it idiosyncratic? This wasn't really mentioned, which seems an oversight.

Response: We have added a simple analysis to address this question. Species relative contributions to R_0 were uncorrelated with mass, and did not differ between native/introduced species, so appear to be idiosyncratic, as they are for some other vector borne pathogens like West Nile virus.

The idea of long-term, low intensity infections is built into many transmission systems. Oft ignored, vector-borne filaria, in large part, adopt this strategy, as do most species of human malaria (the authors reference *P. falciparum*, but it is the other species of malaria, especially *P. vivax* that are the master's of hiding out in a host, right?).

Response: We have added links and references to other systems as suggested.

The part of the discussion that focuses on the specifics of the individual species (house finch—side note, why is this capitalized? Unless it is named after a Dr. House, I think house finch is fine) drifts into areas that are not of general interest.

Response: We have revised and condensed this section, as suggested, to focus on the issues of general interest and have removed the focus on Hawaii-specific details. We would like to keep “House Finch” capitalized (same as Northern Cardinal, House Sparrow, etc.) as that is the convention in ornithology (and the American Ornithological Society), where English and Hawaiian common names are capitalized when referring to specifics of a species.

Methods. No comments, but I do note the tremendous amount of work that went into this study.

Response: We appreciate the acknowledgement!